# Prolonged development of long-term potentiation at lateral entorhinal cortex synapses onto adult-born neurons

Nicholas P. Vyleta, Jason S. Snyder *

Department of Psychology, Djavad Mowafaghian Centre for Brain Health, University of British Columbia, Vancouver, BC, Canada

* jasonsnyder@psych.ubc.ca

**Data Availability Statement:** All relevant data are within the paper and its Supporting Information files.

**Funding:** This work was supported by the Canadian Foundation for Innovation (JSS), the

## Abstract

Critical period plasticity at adult-born neuron synapses is widely believed to contribute to the learning and memory functions of the hippocampus. Experience regulates circuit integration and for a transient interval, until cells are ~6 weeks old, new neurons display enhanced long-term potentiation (LTP) at afferent and efferent synapses. Since neurogenesis declines substantially with age, this raises questions about the extent of lasting plasticity offered by adult-born neurons. Notably, however, the hippocampus receives sensory information from two major cortical pathways. Broadly speaking, the medial entorhinal cortex conveys spatial information to the hippocampus via the medial perforant path (MPP), and the lateral entorhinal cortex, via the lateral perforant path (LPP), codes for the cues and items that make experiences unique. While enhanced critical period plasticity at MPP synapses is relatively well characterized, no studies have examined long-term plasticity at LPP synapses onto adult-born neurons, even though the lateral entorhinal cortex is uniquely vulnerable to aging and Alzheimer's pathology. We therefore investigated LTP at LPP inputs both within (4–6 weeks) and beyond (8+ weeks) the traditional critical period. At immature stages, adult-born neurons did not undergo significant LTP at LPP synapses, and often displayed long-term depression after theta burst stimulation. However, over the course of 3–4 months, adult-born neurons displayed increasingly greater amounts of LTP. Analyses of short-term plasticity point towards a presynaptic mechanism, where transmitter release probability declines as cells mature, providing a greater dynamic range for strengthening synapses. Collectively, our findings identify a novel form of new neuron plasticity that develops over an extended interval, and may therefore be relevant for maintaining cognitive function in aging.

## Introduction

Current theories about the function of adult hippocampal neurogenesis are built upon critical period concepts, where new neurons make important or unique contributions during their immature stages [1–5]. In rodents, adult-born granule neurons begin to form excitatory synapses at ~2 weeks of age and, from this point until they are ~6 weeks old, they have greater

Canadian Institutes of Health Research (JSS) and the Michael Smith Foundation for Health Research (JSS). The funders had no role in study design, data collection and analysis, decision to publish, or preparation of the manuscript.

**Competing interests:** The authors have declared that no competing interests exist.

synaptic plasticity at their afferent [6–9] and efferent synapses [10]. At discrete stages within this window of immaturity, new neurons are more likely to undergo experience-dependent synaptic integration [11–13], morphological remodeling [14] and neuronal survival [15–18]. Given the links between plasticity and memory [19], it is therefore generally believed that new neurons make the greatest contribution to learning during their ~6w critical period, and that their subsequent functional properties are defined by experiences that occurred during immaturity [1–3]. Since cell proliferation declines with age [20–22], there would appear to be a substantial loss of neurogenic plasticity by middle age in mammals.

While adult-born neurons certainly undergo dynamic changes during the first few weeks after cell division, there is emerging evidence that some aspects of neuronal maturation and plasticity may extend beyond the conventional critical period of neuronal development [23]. For example, we recently reported that adult-born neurons in rats continue to grow dendrites and spines, and enlarge their presynaptic terminals from 7–24 weeks of cell age [24]. In conjunction with ongoing low rates of cell addition, we estimated that this extended window of morphological growth could provide the hippocampus with substantial plasticity throughout aging. To date, however, there is no evidence that adult-born neurons go through a similarly extended period of physiological maturation.

The timecourse of new neuron plasticity is particularly relevant from the perspective of aging and cognitive decline. The hippocampus is a major site of convergence of sensory information, where the medial entorhinal cortex axons (the medial perforant path, MPP) broadly conveys spatial information and lateral entorhinal cortex axons (the lateral perforant path, LPP) provides signals about the sensory details that makes each experience unique [25–27]. While it has long been known that the perforant path deteriorates with age in humans [28,29] and animals [30,31], recent evidence suggests that the lateral entorhinal cortex may be particularly vulnerable to age-related tau pathology and functional decline [32–36]. Notably, anatomical and physiological studies indicate that adult-born neurons are preferentially innervated by the LPP [37,38], suggesting neurogenesis may contribute significant plasticity to a vulnerable pathway. However, studies of afferent long-term synaptic plasticity have exclusively focused on the MPP inputs onto adult-born neurons [6–9,39,40].

To gain an understanding of the timecourse of electrophysiological plasticity at a key synapse involved in memory and age-related pathology, we examined long-term potentiation (LTP) at the LPP inputs onto adult-born neurons from 4 to 39 weeks of cell age. In contrast to the critical period that has been described at MPP inputs, we found that LPP LTP increased with cell age over the course several months. These data provide new evidence that adult-born neurons acquire some forms of plasticity over extended intervals, and may provide an important source of synaptic plasticity in the aging brain.

## Methods

### Animals

All procedures were approved by the Animal Care Committee at the University of British Columbia (UBC) and conducted in accordance with the Canadian Council on Animal Care guidelines regarding humane and ethical treatment of animals. Ascl1$^{CreERT2}$ mice (Ascl1$^{tm1.1(Cre/ERT2)Jejo}$; JAX 12882v; [41]) and Ai14 reporter mice (*Gt(ROSA)26Sor$^{tm14(CAG-tdTomato)Hze}$*; JAX 7908; [42]) were purchased from The Jackson Laboratory, and were crossed to generate offspring that were heterozygous for Ascl1$^{CreERT2}$ and homozygous for the Cre-dependent tdTomato reporter, as described elsewhere [43] (hereafter, Ascl1$^{CreERT2}$ mice). Mice were maintained on a C57Bl/6J background, housed 5/cage (floor space 82 square inches), with ad lib access to food and water and a 12hr light-dark schedule with lights on at 7am. To induce

tdTomato expression in Ascl1[+] precursor cells and their progeny, mice were injected intraperitoneally with tamoxifen either neonatally (postnatal day zero or one; ~75 mg/kg, one injection) or during adulthood (6- to 8-weeks-old; 150 mg/kg body weight, one injection/day for up to three days; Fig 1) to permanently label newborn neurons. Adult mice of both sexes were used for electrophysiology experiments between 11- and 45- weeks of age.

## Brain slice preparation

Mice were anesthetized with sodium pentobarbital (intraperitoneal injection, 50 mg/kg) immediately before cardiac perfusion with ice-cold cutting solution containing (in mM): 93 NMDG, 2.5 KCl, 1.2 $NaH_2PO_4$, 30 $NaHCO_3$, 20 HEPES, 25 glucose, 5 sodium ascorbate, 3 sodium pyruvate, 10 n-acetyl cysteine, 0.5 $CaCl_2$, 10 $MgCl_2$ (pH-adjusted to 7.4 with HCl and equilibrated with 95% $O_2$ and 5% $CO_2$, ~310 mOsm). Mice were then decapitated, brains removed, and transverse hippocampal slices prepared as described previously [44]. Slices from the right and/or left hemisphere were transferred to NMDG-containing cutting solution at 35˚C for 20 minutes, before being transferred to a storage solution containing (in mM): 87 NaCl, 25 $NaHCO_3$, 2.5 KCl, 1.25 $NaH_2PO_4$, 10 glucose, 75 sucrose, 0.5 $CaCl_2$, 7 $MgCl_2$ (equilibrated with 95% $O_2$ and 5% $CO_2$, ~325 mOsm) for at least 40 minutes at 35˚C before starting experiments.

## Electrophysiology

Whole-cell patch-clamp recordings were made at near-physiological temperature (~32˚C) from identified tdTomato[+] granule cells in the suprapyramidal blade of the dentate gyrus. Slices were superfused with an artificial cerebrospinal fluid (ACSF) containing (in mM): 125 NaCl, 25 $NaHCO_3$, 2.5 KCl, 1.25 $NaH_2PO_4$, 25 glucose, 1.2 $CaCl_2$, 1 $MgCl_2$ (equilibrated with 95% $O_2$ and 5% $CO_2$, ~320 mOsm). In all experiments GABAergic inhibition was blocked with bicuculline methiodide (10 uM [9]). Recording pipettes were fabricated from 2.0 mm/ 1.16 mM (OD/ID) borosilicate glass capillaries and had resistance ~5 MOhm with an internal solution containing (in mM): 120 K-gluconate, 15 KCl, 2 MgATP, 10 HEPES, 0.1 EGTA, 0.3 $Na_2GTP$, 7 $Na_2$-phosphocreatine (pH 7.28 with KOH, ~300 mOsm). Current-clamp and voltage-clamp recordings were performed at -80 mV. Only recordings with high seal resistance (several giga-ohms) and low holding current (less than 50 pA) were included in analyses. For current-clamp recordings, series resistance and pipette capacitance were compensated with the bridge balance and capacitance neutralization circuits of the amplifier. A bipolar electrode was placed in the outer 1/3 of the molecular layer to stimulate the lateral perforant path (LPP) fibers ([45,46]; Fig 1B). Stimuli (0.1 ms) were delivered through a stimulus isolator (A-M Systems analog stimulus isolator model 2200) and intensity (range 50–500 μA, median 200 μA; did not differ with cell age, correlation P = 0.95) was adjusted to evoke minimum excitatory postsynaptic currents (EPSCs; -40 ± 4 pA, mean ± standard error (here and elsewhere)) and corresponding excitatory postsynaptic potentials (EPSPs) ~5 mV (5.2 ± 0.5 mV). Paired-pulse facilitation was assessed using 50-Hz pairs of pulses. For LTP experiments, single EPSPs were evoked every thirty seconds before and after a single theta-burst stimulation (TBS) consisting of 10 trains of 10 pulses (100-Hz), delivered at 5-Hz, and repeated four times at 0.1 Hz, paired with postsynaptic current injection (100 pA, 100 ms) as previously described [8,9].

## Data acquisition and analysis

Data were acquired with a Multiclamp 700B amplifier, low-pass filtered at 10 kHz, and digitized at 100 kHz with an Axon 1550B digitizer. Pulse generation and data acquisition were performed using pClamp 10 (Molecular Devices). EPSC and EPSP traces were analyzed offline

Fig 1. Recording from neonatal and adult-born dentate granule neurons. (A) Timelines for labelling and recording from neonatal- and adult-born dentate granule neurons. (B) Fluorescence (left) and IR-DIC (middle) images of a tdTomato+ adult-born granule cell (39 days post-tamoxifen injection) that was targeted for whole-cell recording. The right panel shows the low magnification view, where the stimulating electrode is placed in the outer molecular layer to target the lateral perforant path axons that arise from the lateral entorhinal cortex (gcl, granule cell layer; hil, hilus; mol, molecular layer). (C) Input resistance declines with time post-tamoxifen, consistent with the known age-related

physiological maturation of adult-born granule cells ($R^2 = 0.37$, P < 0.0001). (**D**) Young adult-born granule cells had higher input resistance than older adult-born or neonatal-born cells (Kruskal Wallis test, P < 0.0001; 4-6w vs 8+w, ****P < 0.0001; 4-6w vs neonatal, *P = 0.01; 8+w vs neonatal, P = 0.5). Bars reflect mean ± standard error.

using Clampfit (Molecular Devices) and Igor Pro (Wavemetrics) software. Input resistance was measured from a test pulse (10 mV) in voltage-clamp. Peak EPSC amplitudes were measured from average waveforms of 10 consecutive traces collected at 0.1 Hz, and from a baseline period immediately preceding each stimulus. LTP magnitude was measured as the mean peak EPSP amplitude during 40–50 minutes post-TBS normalized to the mean peak EPSP amplitude during ten minutes of baseline recording immediately preceding the TBS. Paired-pulse responses were collected immediately before and after each LTP experiment. Paired pulse ratios were calculated as the peak EPSC amplitude of the 2nd response divided by the peak EPSC amplitude of the 1st response. For some analyses, adult-born neurons were grouped into bins of 4–6 weeks and 8+ weeks post-tamoxifen injection, to specifically compare cohorts of cells that are within and beyond, respectively, the critical period for LTP at medial perforant path synapses [9]. Individual data points reflect cells; only 1 cell was examined per slice and 1–2 cells were examined per animal. Since no differences were observed between adult-born cells from male vs female mice (LTP, input resistance and paired pulse ratio all P > 0.26), data from both sexes were pooled for all analyses. Total number of cells analyzed: 4-6w adult-born cells, n = 11; 8+w adult-born cells, n = 26; neonatal-born cells, n = 11, with the exception that sample sizes were slightly smaller for post-TBS paired pulse ratios: 4-6w adult-born cells, n = 8; 8+w adult-born cells, n = 22; neonatal-born cells, n = 10. Group data are expressed as means ± standard error.

Cell age-related physiological differences were analyzed by regression and group differences were identified by ANOVA and followed up with Holm-Sidak comparison tests. If data were non-normal, group differences were identified by a Kruskal-Wallis test with Dunn's post-hoc test. Changes in paired pulse ratios were analyzed by t-test or, if the data were not normally distributed, Mann Whitney test. To facilitate comparison with data presented in graphs, most statistical analyses are described in the figure legends. For all analyses, statistical significance was defined as P < 0.05. The data for all graphs and analyses are provided as supporting information (S1 File).

## Results

We investigated long-term potentiation (LTP) of synaptic transmission at LPP synapses onto immature and mature adult-born dentate granule cells. We used Ascl1^CreERT2 mice, where tamoxifen injection labels Ascl1+ precursor cells and their neuronal progeny with a tdTomato reporter [41] (Fig 1A). While tamoxifen labels Ascl1+ precursor cells that may divide at later dates, Ascl1+ cells are typically non-renewing and produce the majority of their neuronal daughter cells within ~2–3 weeks after tamoxifen injection [47,48]. Consistent with relatively precise birthdating, the timecourse of electrophysiological maturation following tamoxifen injection closely parallels that of retrovirally-labelled adult-born granule cells [43]. Nonetheless, to confirm this, we performed whole-cell patch-clamp recordings and measured cellular input resistance, which reliably declines as granule cells mature (i.e. grow in size and express inwardly rectifying K+ channels [49,50]). Indeed, when tamoxifen was administered in adulthood, input resistance was negatively correlated with the post-injection interval (Fig 1C), and young adult-born granule cells (4–6 weeks) had greater input resistance than mature adult-born granule cells (8+ weeks) and cells labelled by neonatal tamoxifen injection (Fig 1D). In contrast, input resistance did not correlate with the post-injection interval when tamoxifen

was given neonatally (suggesting that, at the time of recording, 101–173 days later, these cells were fully mature; P = 0.14). These data indicate that tamoxifen effectively labels distinct cohorts of physiologically immature and mature granule cells over time.

Synaptic transmission was monitored by recording excitatory postsynaptic potentials (EPSPs) in current-clamp configuration following low-frequency stimulation (every 30 seconds) of the LPP. Baseline EPSP, but not EPSC, amplitudes negatively correlated with the post-tamoxifen interval (S1 Fig). This is most likely due to the cell age-related decline in input resistance (Fig 1C), since $\Delta V = IR$. To evoke long-term synaptic plasticity, we used an established theta-burst stimulation (TBS) paradigm that elicits LTP at medial perforant path inputs onto immature granule cells [8,9]. TBS resulted in both LTP and long-term depression (LTD) at LPP synapses onto adult-born granule cells (Fig 2). As a group, younger 4-6w cells did not undergo significant LTP (Wilcoxon signed rank test, P = 0.5) and more frequently underwent long-term depression (5/11 cells vs 4/26 cells at 8[+] weeks). In contrast, older adult-born cells and neonatal-born cells did exhibit significant LTP (8[+]-week-old cells, P < 0.0001; neonatal cells, P = 0.01). Notably, 8[+]w cells underwent greater potentiation than 4-6w cells (300% vs.

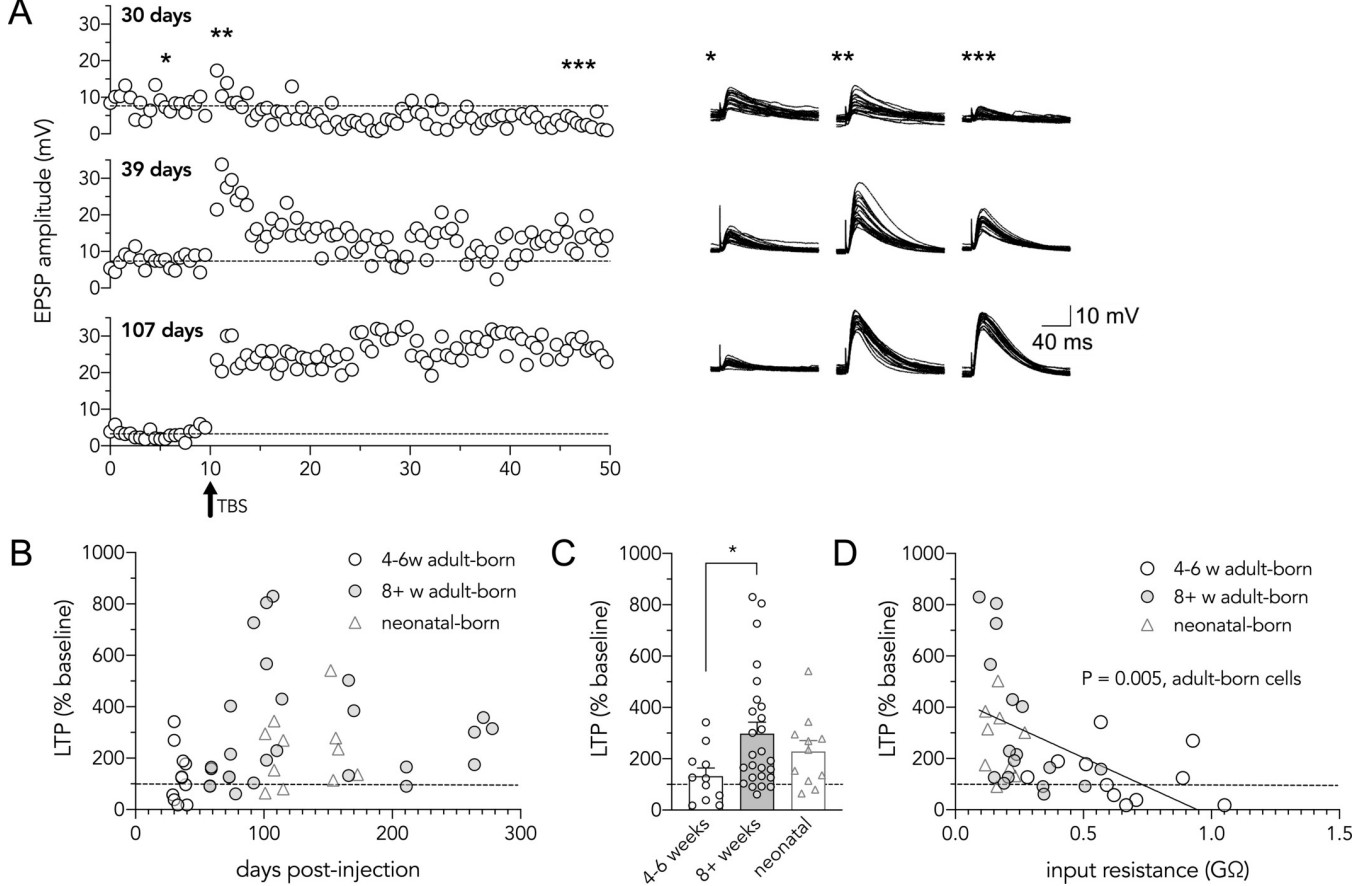

**Fig 2. Lateral perforant path LTP increases with adult-born neuron age.** (**A**) Left: Representative plots of peak excitatory postsynaptic potential (EPSP) amplitude as a function of time for three different adult-born dentate granule cells (30-, 39-, and 107 days post-injection). Single EPSPs (lateral perforant path) were evoked every 30 seconds before and after a single theta-burst stimulation (delivered after ten minutes of baseline recording). Right: Single EPSPs overlaid during baseline recording, immediately following TBS, and during 40–50 minutes of recording (30 minutes post-TBS). (**B**) Long-term potentiation (LTP) as a function of days post-injection for adult- and neonatal-born dentate granule cells. (**C**) Mature adult-born cells underwent greater LTP than immature cells and did not differ from neonatal-born cells (Kruskal Wallis test, P < 0.05; 4-6w cells vs 8[+]w cells, *P = 0.03; 8[+]w cells vs neonatal cells, P = 0.99). (**D**) Long-term potentiation plotted as a function of input resistance for adult-born and neonatal-born granule cells. More mature (lower input resistance) adult-born granule cells have greater LTP ($R^2$ = 0.27, P = 0.005). Bars reflect mean ± standard error.

130%, respectively; Fig 2C). As a group, older adult-born neuron LTP did not differ from neonatal neurons, but a subset of ~15 week-old adult-born cells displayed the greatest amount of LTP (~600–800%; Fig 2B). Finally, LTP was inversely correlated with the input resistance of adult-born cells, confirming that the LPP undergoes stronger potentiation at synapses onto more mature granule cells (Fig 2D). LTP did not correlate with input resistance among neonatal-born cells ($R^2 = 0.04$, P = 0.6).

While LTP at LPP synapses is induced postsynaptically via NMDA receptors, it is ultimately expressed through an increased probability of transmitter release [51,52]. We therefore investigated whether LPP LTP, in our hands, displayed presynaptic characteristics. Short-term synaptic plasticity was measured by recording excitatory postsynaptic currents (EPSCs) evoked by paired pulse stimulation of LPP afferents (50 Hz) in voltage clamp (Fig 3). All recordings showed paired-pulse facilitation, where the second EPSC was greater than the first (paired pulse ratio, PPR, > 1). This form of short-term plasticity reflects an increased probability of neurotransmitter release [53], and is well-established at LPP-granule cell synapses [46,51,52]. In line with previous reports, there was a significant reduction in paired-pulse facilitation following TBS, consistent with a presynaptic mechanism whereby LPP synapses potentiate via an increase in release probability [51,52] (Fig 3A; PPR = 2.3 at baseline vs. 1.8 after LTP; all adult-born and neonatal-born cells pooled, but data for all groups available as supporting information). Importantly, the magnitude of LTP was predicted by both initial (baseline) facilitation as

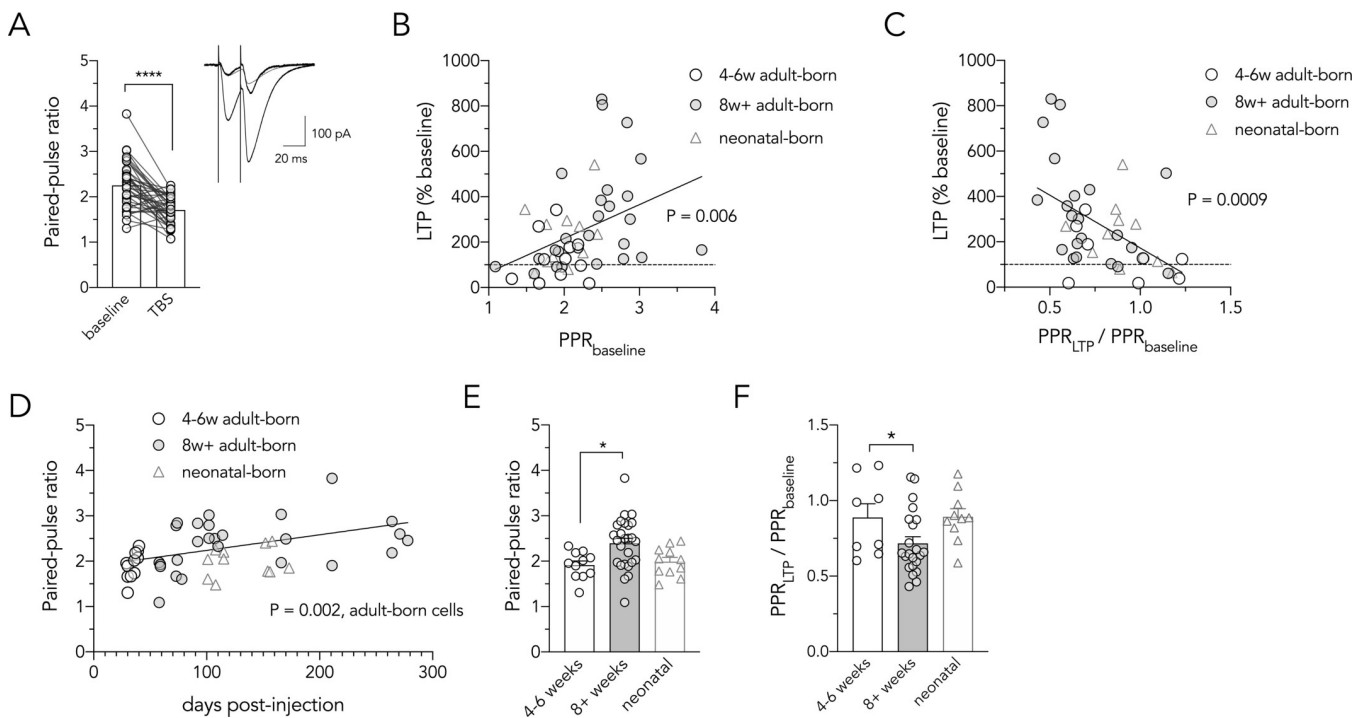

**Fig 3. Older adult-born neurons have greater LTP due to presynaptic plasticity.** (**A**) TBS reduced the paired-pulse ratio of excitatory postsynaptic currents at most LPP-to-granule cell synapses (all adult-born and neonatal-born cells pooled; $T_{39} = 6.1$, P < 0.0001). Inset shows pairs of EPSCs recorded from an adult-born granule cell (107 DPI) before and after TBS-induced LTP (black traces). Normalizing the potentiated response to the peak of the first baseline EPSC (grey trace) illustrates the reduction in paired-pulse facilitation. (**B**) Paired-pulse ratio at baseline (before TBS) correlates with subsequent LTP magnitude (all adult-born and neonatal-born cells pooled; $R^2 = 0.15$, P = 0.006). (**C**) TBS-induced reduction in paired-pulse ratio correlates with the magnitude of LTP at granule cell synapses (all adult-born and neonatal-born cells pooled; $R^2 = 0.25$, P = 0.0009). (**D**) Paired-pulse ratio increases with the age of the adult-born granule cell ($R^2 = 0.23$, P = 0.002). (**E**) Baseline paired pulse ratio was greater for mature adult-born granule cells ($T_{35} = 2.6$, P = 0.01). (**F**) Older adult-born neurons underwent a greater reduction in paired-pulse facilitation following LTP (Mann Whitney test, P = 0.03). *P < 0.05, ****P < 0.0001). Bars reflect mean ± standard error.

well as by the magnitude of the reduction in facilitation (and thus increase in release probability) after LTP (Fig 3B and 3C). Thus, synapses with a lower initial release probability could undergo greater enhancement of neurotransmitter release during LTP.

We next examined whether presynaptic physiology differs as a function of adult-born cell age. Indeed, paired-pulse facilitation increased with the age of postsynaptic adult-born granule cell (but not neonatal-born cells: $R^2 = 0.03$, $P = 0.6$; Fig 3D), and was greater for mature than for immature adult-born cells (PPR in 8w+ cells = 2.4, PPR in 4-6w cells = 1.9; Fig 3E). These data indicate that LPP inputs have a low initial probability of transmitter release onto mature adult-born cells. To investigate whether enhanced release underlies the greater LTP in older adult-born neurons, we compared PPR changes in 4-6w cells and 8+w cells and found that, indeed, inputs onto more mature cells displayed a greater reduction in facilitation after TBS (i.e. greater enhancement of release probability; Fig 3F). Taken together, these data suggest that synapses between LEC neurons and adult-born granule cells mature with age, which reduces release probability and enables synapses onto older neurons to realize stronger LTP.

## Discussion

Here we report a novel, age-related pattern of long-term plasticity at cortical input synapses onto adult-born hippocampal neurons. Whereas LTP at MPP synapses is greatest when adult-born neurons are in an immature critical period, here we found that immature cells do not reliably potentiate at LPP synapses but instead develop increasingly greater capacity for LTP with age and cellular maturity–both in terms of magnitude of LTP and percent of cells undergoing potentiation. Given the distinct roles of the medial and lateral entorhinal cortices in memory, and their vulnerability to age-related pathology, neurogenesis may therefore make a unique and important contribution to hippocampal cognition in adulthood and aging.

### Old adult-born neurons have greater LTP at lateral perforant path synapses

The majority of studies of DG LTP, both within and beyond the field of adult neurogenesis, have focussed on plasticity at MPP synapses. With respect to neurogenesis, one of the most consistent findings is the enhanced LTP at MPP synapses onto immature adult-born neurons, which has been demonstrated in mice and rats, using radiological [7], chemical [39] and transgenic [40] methods to inhibit (or enhance [54]) neurogenesis, and has been directly verified with whole cell recordings from immature [6,8] and birthdated [9] neurons. In contrast, little is known about the physiologically and pharmacologically distinct LPP pathway, though recent reports indicate striking differences between LPP and MPP innervation of adult-born neurons. Whereas DG granule cells are widely understood to receive relatively equal innervation from both the LEC and MEC, immature adult-born neurons are primarily targeted by the LPP [37,38], though innervation from both pathways can further increase with age and experience [12,55]. While many aspects of the synaptic physiology of 7-week-old adult-born neurons are comparable to neonatal-born neurons [56,57], our results identify a form of long-term synaptic plasticity that matures over several months.

What is the mechanism of enhanced LTP in older adult-born neurons? A recent report demonstrated that induction of LTP at LPP–granule cell synapses is dependent on postsynaptic NMDARs and metabotropic glutamate receptors, but expression is mediated through activation of cannabinoid receptors (CB1) on the presynaptic terminals and enhancement of release probability [52]. Our results are consistent with this presynaptic expression, and we show that synapses onto older adult-born granule cells have reduced release probability at baseline (more facilitation), and thus have a greater dynamic range for enhancement upon

LTP. Reduced release probability and enhanced facilitation at older synapses could result from increased presynaptic calcium buffers or longer coupling distances between calcium channels and synaptic vesicles [58] or by reliance on different subtypes of presynaptic calcium channels [59]. Potentiation of release following TBS may occur via increase in the number of calcium channels at presynaptic active zones [60] or by other mechanisms. The non-canonical endo-cannabinoid signaling pathway, whereby activation of CB1 receptors *increases* transmitter release [52] instead of the more typical reduction seen at other neuronal pathways [61], may explain the occurrence of both LTP and LTD in the current experiments (Fig 2). Possibly, CB1 receptors either reduce or enhance transmitter release depending on the maturity of the synapse if, for example, immature synapses have low CB1 receptor activation and mature synapses have high CB1 receptor activation [62]. Notably, the transient enhancement of EPSP amplitude immediately following the TBS (Fig 3A), similar to post-tetanic potentiation (PTP), did not depend on the age of adult-born granule cell (correlation, P = 0.96; 8w$^+$ cells = 3.2 ± 0.5, 4-6w cells = 2.8 ± 0.5; t-test, P = 0.6). Based on recent work demonstrating that PTP results from a transient enhancement of the readily releasable pool of synaptic vesicles [63], we suggest that vesicle pool enlargement does not underlie the greater LTP observed here at older granule cells. Importantly, we observed no differences in postsynaptic spiking during LTP induction (correlation between total spikes and LTP magnitude, P = 0.6), suggesting that differences in granule cell activity do not explain our findings, though differences in dendritic spiking [8,64,65] or other postsynaptic signals may contribute to presynaptic LTP expression.

Conclusions about age-related plasticity depend on the methods used to birthdate neurons. Here, we used Ascl1$^{CreERT2}$ mice, where tamoxifen injection labels Ascl1$^+$ precursor cells that may divide immediately after injection or after a delay [41]. Cellular birthdating is therefore not as precise as with retroviral vectors, which only label actively dividing cells. However, our central finding, that LPP LTP increases with cell age, is largely unaffected by this limitation for several reasons. First, modelling the timecourse of neuronal maturation in Ascl1$^{CreERT2}$ mice suggests that tamoxifen labels a cohort of cells that are largely born around the time of injection [43]. Recent in vivo imaging of Ascl1$^{CreERT2}$ mice confirms this, and has indicated that these cells are non-renewing, divide by ~12 days post-injection, and produce the majority of their daughter cells within ~10 days of division [48]. Thus, while there may be some loss of temporal resolution, the majority of cells are generated in a window of time that is much smaller than the timecourse of LTP changes we observed here. Furthermore, the delayed division of Ascl1$^+$ cells would result in cells that may be 1–3 weeks younger than "days post-injection". Based on previous results from MPP synapses onto retrovirally labelled neurons [9], we would then expect our younger cells, in the 4-6w group, to reliably undergo LTP and yet we consistently observed no potentiation or even LTD. The second major line of evidence supporting our interpretation is the fact that LTP strongly correlated with input resistance, an independent and well-established physiological measure of cell maturity in the developing [49] and adult [8,50] DG. In fact, LTP more strongly correlated with input resistance than days post-injection (likely due to the lower temporal precision of the latter). For these reasons, the most likely interpretation of our data is that LPP LTP is weak in immature cells and progressively increases over 3–4 months as newborn neurons mature.

## Implications for cognition and aging

Critical period properties are central to many theories about the function of adult neurogenesis [1–4]. Broadly speaking, transient windows of enhanced synaptic plasticity are thought to make new neurons particularly sensitive to sensory inputs arriving from the entorhinal cortex. In this way, a major contribution to learning, or the tuning of their receptive field properties,

occurs during their immature stages of development. Given that neurogenesis declines by 90% from young adulthood to middle age [20,21,66], it might appear that neurogenesis has little to offer later in life. Neurogenesis may still make important contributions later in aging, through cumulative cell addition, and the possibility that functionally distinct cells are produced in adulthood vs development [23,67]. However, the timecourse of development is also an important factor to consider. For example, we have recently found that adult-born neurons in rats continue to grow dendrites, spines, and presynaptic terminals over 6 months which, cumulatively, results in substantial morphological plasticity in aging, even after cell proliferation has declined to low levels [24]. Our current results identify a form of physiological plasticity that also develops over an extended timeframe, is robust in older neurons, and may therefore facilitate learning in the aged brain.

How might LPP LTP contribute to specific behavioral processes? Whereas MEC cells code for space [68] and movement [69], LEC cells have been found to respond to specific cues, such as objects [70,71] and odors [72,73]. Lesion studies also broadly implicate the MEC in spatial memory and the LEC in object-related memory [74–77]. These approximate divisions of labor reflect upstream inputs from the dorsal and ventral processing streams. Convergence of signals coding for spatial context (MEC) and sensory content (LEC) then leads to precise, experience-specific representations in the hippocampus [25,78]. Preferential targeting by the LEC [37,38], and the extended development of LEC–new neuron plasticity reported here, suggests that adult-born neurons may especially facilitate learning about the cues that make each experience unique. Such a function could contribute to learning about, or responding to, discrete objects and cues [79–82] (but see [83]). Roles for LEC in learning about cue configurations may also underlie new neuron functions in discrimination between similar contexts and places [39,54,84,85]. It is less clear how afferent LEC plasticity contributes to the non-mnemonic functions of neurogenesis [86], such as stress responding and anxiety [87–89], but the entorhinal cortex does regulate defensive behaviors in primates [90], and has extensive connectivity with the amygdala [91]. Given the unusually rich connectivity of the LEC with other brain regions [92], extended plasticity at adult-born neuron synapses may have broad implications for memory and behavior regulation.

Plasticity at the LPP-DG synapse is particularly relevant for cognitive aging given convergent evidence for entorhinal, and specifically LEC vulnerability in aging and Alzheimer's disease. Indeed, the perforant path is sensitive to age related pathology [28,29] and LEC-related object memory deteriorates in aging prior to more global deficits or clinical diagnoses [33,35,93,94]. Likewise, in rats, object discrimination declines with age and is associated with abnormal patterns of LEC activity [95,96] and LPP LTP is reduced as early as 6 months of age in mice [36]. Our results suggest that adult-born neurons may provide a valuable source of plasticity to a highly vulnerable circuit, and may be a relevant target for promoting LEC-related behavioral functions later in life. While our binned analyses suggest that LTP in old adult-born neurons is comparable to that of neonatal-born neurons, our groupings spanned large age ranges (both for cell age and animal age) and so additional study is warranted. For example, the greatest amount of LTP was observed in a handful of adult-born cells at ~15 weeks post-injection, which suggests a possible delayed critical period. Alternatively, there may be an inverted U-relationship between cell age and LTP magnitude, where the ascending phase reflects cellular maturation and the declining phase reflects a more general (animal level) age-related decline in LPP LTP, which is already apparent by 6 months [36]. This may have therefore led to a reduction in LTP magnitude selectively in our mature adult-born group, since some of these recordings came from older animals. Given that mature neonatal-born neurons are more vulnerable to delayed cell death [97–99], it will be important to examine older animals and determine whether they are also more susceptible to age-related synaptic deterioration.

## Supporting information

**S1 Fig. Baseline synaptic transmission.** A) Baseline peak EPSP amplitudes did not significantly differ across groups (Kruskal Wallis test, P = 0.09). Among adult-born cells, EPSP amplitude negatively correlated with days post-tamoxifen injection ($R^2$ = 0.23, P = 0.0025). C) Among adult-born cells, EPSC amplitude did not vary across groups (Kruskal Wallis test, P = 0.9). D) EPSC amplitude did not correlate with days post-tamoxifen injection ($R^2$ = 0.05, P = 0.18). Bars reflect mean ± standard error.
(JPG)

**S1 File. Underlying data for all analyses.**
(XLSX)

## Author Contributions

**Conceptualization:** Nicholas P. Vyleta, Jason S. Snyder.

**Formal analysis:** Nicholas P. Vyleta, Jason S. Snyder.

**Funding acquisition:** Jason S. Snyder.

**Methodology:** Nicholas P. Vyleta.

**Writing – original draft:** Nicholas P. Vyleta.

**Writing – review & editing:** Nicholas P. Vyleta, Jason S. Snyder.

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
