## [Decision Letter · Decision Letter 0]

6 May 2021

PONE-D-21-08869

Prolonged development of long-term potentiation at lateral entorhinal cortex synapses onto adult-born neurons

PLOS ONE

Dear Dr. Snyder,

Thank you for submitting your manuscript to PLOS ONE. After careful consideration, we feel that it has merit but does not fully meet PLOS ONE’s publication criteria as it currently stands. Therefore, we invite you to submit a revised version of the manuscript that addresses the points raised during the review process.

We look forward to receiving your revised manuscript.

Kind regards,

Brian R Christie

Academic Editor

PLOS ONE

Additional Editor Comments:

Hi Jason,

Both reviewers liked the work, but had some issues with the figures and some of the PPR data primarily. They seem to convey pretty clearly their concerns, so they should be straightforward changes to make.

bri

Journal Requirements:

Reviewers' comments:

Reviewer's Responses to Questions

**Comments to the Author**

1. Is the manuscript technically sound, and do the data support the conclusions?

Reviewer #1: Yes

Reviewer #2: Yes

2. Has the statistical analysis been performed appropriately and rigorously? 

Reviewer #1: Yes

Reviewer #2: Yes

3. Have the authors made all data underlying the findings in their manuscript fully available?

Reviewer #1: Yes

Reviewer #2: Yes

4. Is the manuscript presented in an intelligible fashion and written in standard English?

Reviewer #1: Yes

Reviewer #2: Yes

5. Review Comments to the Author

Reviewer #1: In this study the authors examined long term synaptic plasticity in newly born neurons from neonates, adults 4-6 weeks and adults 8+ weeks old. They found differing amounts of long-term plasticity that positively correlated to a decrease in probability of neurotransmitter release. Overall this is an interesting study but some issues need to be addressed.

Major Issues

-I assume that experiments were performed under an approved animal protocol. Because this was not stated in the Methods. Please state if this is the case in the Methods.

-In Figure 2 it appears from Fig2A that the amplitudes of the baseline EPSPs progressively get smaller with age. There should be a summary graph comparing baseline amplitudes for neonatal, adult 4-6 weeks and adult 8+ weeks.

-For Fig 2C we have three groups. 4-6 weeks adult; 8+ weeks adult; and neonatal.

These three groups should also be indicated in Figs 2B and D as circles of different shades. For example, neonatals can be grey fill circles, adult 4-6 weeks as white circles and adult 8+ weeks as black circles or whatever shows best.

-For Figure 3, it would be interesting to add the paired pulse data for neonatal mice and also indicate the different ages in the correlation graphs with different shades of circles.

Minor Issue

-Page 3 Line 6 of Methods: Do you actually mean hemizygous or do you mean heterozygous?

Reviewer #2: This is a nicely done short report investigating whether LPP synapses onto adult born neurons can express NMDAR-dependent LTP. The data appear to be of high quality and the manuscript is for the most part clearly written. The conclusions are aligned with the findings. There are some edits needed that can improve the manuscript.

1) The y axis label for all of the figures currently with the "LTP" label (e.g. Fig 1B-D) is confusing. What do the numbers, 0-10 represent? Shouldn't it be percent of baseline, or percent potentiation? As currently labeled, it is not possible to know what is being shown.

2) The single representative examples of LTP shown in Fig 1A are nice, but it would also have been nice to see the averaged LTP data plotted over time for each group as well.

3) The PPR data shown in Fig 3A is only from the "older" adult born neurons. It would be helpful to see the same data from the "younger" ones too, before and after TBS, even though LTP expression did not occur. This is sort of shown in Fig 3F with the data plotted as a ratio, but it would be helpful to also see it plotted as in Fig 3A.

4) Might be clearer to say that the magnitude of LTP rather than extent of LTP. It is difficult to know what is meant by "extent"- could be magnitude or could be how long-lasting the potentiation is (e.g. early LTP vs late, protein synthesis dependent LTP). When the authors say, "...increasingly greater capacity for LTP with age and cellular maturity" do they mean a greater magnitude of LTP, or a greater percentage of the time LTP is successfully induced and expressed? Being more specific would be helpful. Care should also be taken to make sure induction is not confused with expression when discussing LTP. For example, in the sentence, page 6 line 8, don't the authors really mean expression of LTP rather than induction of LTP?

6. PLOS authors have the option to publish the peer review history of their article (what does this mean?). If published, this will include your full peer review and any attached files.

Reviewer #1: No

Reviewer #2: No

---

## [Author Response · Author response to Decision Letter 0]

17 May 2021

Please find our responses to each point below, in bold.

Reviewer #1: In this study the authors examined long term synaptic plasticity in newly born neurons from neonates, adults 4-6 weeks and adults 8+ weeks old. They found differing amounts of long-term plasticity that positively correlated to a decrease in probability of neurotransmitter release. Overall this is an interesting study but some issues need to be addressed.

Major Issues

-I assume that experiments were performed under an approved animal protocol. Because this was not stated in the Methods. Please state if this is the case in the Methods.

Ah, yes – we negelected to include this and have now added a statement confirming these details (first sentence of the methods section).

-In Figure 2 it appears from Fig2A that the amplitudes of the baseline EPSPs progressively get smaller with age. There should be a summary graph comparing baseline amplitudes for neonatal, adult 4-6 weeks and adult 8+ weeks.

This is a good observation. Indeed, baseline EPSPs did get smaller with increasing cell age. This is most likely because cell input resistance is declining with cell age (Fig. 1C), since we find that the EPSCs do not change over time. Since these data are somewhat peripheral to our main findings, we have included these graphs as supporting information (Fig. S1) and refer to them in the 2nd paragraph of the results section.

-For Fig 2C we have three groups. 4-6 weeks adult; 8+ weeks adult; and neonatal.

These three groups should also be indicated in Figs 2B and D as circles of different shades. For example, neonatals can be grey fill circles, adult 4-6 weeks as white circles and adult 8+ weeks as black circles or whatever shows best.

This is a good idea. Here and elsewhere we have therefore changed the symbols for the 3 groups in our correlational analyses.

-For Figure 3, it would be interesting to add the paired pulse data for neonatal mice and also indicate the different ages in the correlation graphs with different shades of circles. 

We definitely appreciate that it could be interesting to also see the group breakdown for the PPR data in Figure 3 A. However, our main objective with Fig 3A-C was to establish whether, in our hands, LPP LTP appears presynaptic, as others have found. In revising this figure we therefore began to create multiple graphs in Fig 3A for each of the 3 groups. And, while they all showed the same trend, not all were statistically significant, likely due to smaller sample sizes for the 4-6w and neonatal groups. We would therefore prefer to keep this panel simple to illustrate the main point (that, as a whole, LPP LTP at granule cell synapses, is associated with presynaptic changes). We have, however, added mention in the text of the supplementary underlying data, should readers wish to explore the different groups. Also, we have followed the other suggestion made here and changed the symbols accordingly.

Minor Issue

-Page 3 Line 6 of Methods: Do you actually mean hemizygous or do you mean heterozygous?

Indeed we meant heterozygous and have changed the text accordingly.

Reviewer #2: This is a nicely done short report investigating whether LPP synapses onto adult born neurons can express NMDAR-dependent LTP. The data appear to be of high quality and the manuscript is for the most part clearly written. The conclusions are aligned with the findings. There are some edits needed that can improve the manuscript.

1) The y axis label for all of the figures currently with the "LTP" label (e.g. Fig 1B-D) is confusing. What do the numbers, 0-10 represent? Shouldn't it be percent of baseline, or percent potentiation? As currently labeled, it is not possible to know what is being shown.

By LTP we meant “fold potentiation), where 1 = no change. But since this was not clear we have changed the y-axes of all LTP graphs to “LTP (% baseline)”.

2) The single representative examples of LTP shown in Fig 1A are nice, but it would also have been nice to see the averaged LTP data plotted over time for each group as well.

We certainly agree that average time course data can be a useful visualization. However in this case there is so much variability that the averaged data obscures some of the important details. In this case we wished to emphasize the variability that can occur even amongst cells of a similar age. For example the 30 day and 39 day examples are both within the 4 to 6 week group and yet one shows LTD and the other shows LTP. Combining these in one plot versus time could give a false impression of an average phenomenon—in essence it would appear that cells in the 4 to 6 week group showed no change in EPSP amplitude—when in fact most cells showed one of two distinct processes (LTD or LTP) of substantial magnitude. Of course we continue to provide the averaged data for the final LTP magnitude in figure 2C, and provide the underlying data as supplementary material should readers wish to explore the data further.

3) The PPR data shown in Fig 3A is only from the "older" adult born neurons. It would be helpful to see the same data from the "younger" ones too, before and after TBS, even though LTP expression did not occur. This is sort of shown in Fig 3F with the data plotted as a ratio, but it would be helpful to also see it plotted as in Fig 3A.

This figure actually pooled data from all cells regardless of when tamoxifen was injected. We have clarified this in the figure legend, and prefer to keep the data pooled in order to make the simple point that TBS leads to changes in PPR (as discussed above in response to reviewer #1).

4) Might be clearer to say that the magnitude of LTP rather than extent of LTP. It is difficult to know what is meant by "extent"- could be magnitude or could be how long-lasting the potentiation is (e.g. early LTP vs late, protein synthesis dependent LTP). 

We agree and have changed “extent” to “magnitude”

When the authors say, "...increasingly greater capacity for LTP with age and cellular maturity" do they mean a greater magnitude of LTP, or a greater percentage of the time LTP is successfully induced and expressed? Being more specific would be helpful. 

We have changed the text to clarify: “increasingly greater capacity for LTP with age and cellular maturity - both in terms of magnitude of LTP and % of cells undergoing potentiation”

Care should also be taken to make sure induction is not confused with expression when discussing LTP. For example, in the sentence, page 6 line 8, don't the authors really mean expression of LTP rather than induction of LTP?

To avoid any confusion about the stage of LTP we have chaned this to “during” LTP, in addition to the following changes: Page 6, line 2: removed words “induction of”—now reads: “…after LTP.” Page 6, line 48: removed word “induction”—now reads: “…upon LTP. Page 11, line 12 (Fig. 3 Legend): removed word “induction”—now reads: “…following LTP.”

---

## [Decision Letter · Decision Letter 1]

10 Jun 2021

Prolonged development of long-term potentiation at lateral entorhinal cortex synapses onto adult-born neurons

PONE-D-21-08869R1

Dear Dr. Snyder,

We’re pleased to inform you that your manuscript has been judged scientifically suitable for publication and will be formally accepted for publication once it meets all outstanding technical requirements.

Kind regards,

Brian R Christie

Academic Editor

PLOS ONE

Additional Editor Comments (optional):

Reviewers' comments:

Reviewer's Responses to Questions

**Comments to the Author**

1. If the authors have adequately addressed your comments raised in a previous round of review and you feel that this manuscript is now acceptable for publication, you may indicate that here to bypass the “Comments to the Author” section, enter your conflict of interest statement in the “Confidential to Editor” section, and submit your "Accept" recommendation.

Reviewer #1: All comments have been addressed

Reviewer #2: All comments have been addressed

2. Is the manuscript technically sound, and do the data support the conclusions?

Reviewer #1: Yes

Reviewer #2: Yes

3. Has the statistical analysis been performed appropriately and rigorously? 

Reviewer #1: Yes

Reviewer #2: Yes

4. Have the authors made all data underlying the findings in their manuscript fully available?

Reviewer #1: Yes

Reviewer #2: Yes

5. Is the manuscript presented in an intelligible fashion and written in standard English?

Reviewer #1: Yes

Reviewer #2: Yes

6. Review Comments to the Author

Reviewer #1: (No Response)

Reviewer #2: (No Response)

7. PLOS authors have the option to publish the peer review history of their article (what does this mean?). If published, this will include your full peer review and any attached files.

Reviewer #1: No

Reviewer #2: No

---

## [Editor Report · Acceptance letter]

11 Jun 2021

PONE-D-21-08869R1 

­­Prolonged development of long-term potentiation at lateral entorhinal cortex synapses onto adult-born neurons 

Dear Dr. Snyder:

I'm pleased to inform you that your manuscript has been deemed suitable for publication in PLOS ONE. Congratulations! Your manuscript is now with our production department. 

Kind regards, 

on behalf of

Dr. Brian R Christie 

Academic Editor

PLOS ONE